# Regulatory T Cells in the Pathogenesis of Graves’ Disease

**DOI:** 10.3390/ijms242216432

**Published:** 2023-11-17

**Authors:** Natasa Kustrimovic, Daniela Gallo, Eliana Piantanida, Luigi Bartalena, Adriana Lai, Nicola Zerbinati, Maria Laura Tanda, Lorenzo Mortara

**Affiliations:** 1Center for Translational Research on Autoimmune and Allergic Disease—CAAD, Università del Piemonte Orientale, 28100 Novara, Italy; 2Endocrine Unit, Department of Medicine and Surgery, University of Insubria, ASST dei Sette Laghi, 21100 Varese, Italymaria.tanda@uninsubria.it (M.L.T.); 3Dermatology Unit, Department of Medicine and Surgery, University of Insubria, ASST dei Sette Laghi, 21100 Varese, Italy; 4Immunology and General Pathology Laboratory, Department of Biotechnology and Life Sciences, University of Insubria, 21100 Varese, Italy

**Keywords:** Graves’ disease, T regulatory cells, autoimmunity, cellular immunity, T helper 17

## Abstract

Maintaining a delicate balance between the prompt immune response to pathogens and tolerance towards self-antigens and commensals is crucial for health. T regulatory (Treg) cells are pivotal in preserving self-tolerance, serving as negative regulators of inflammation through the secretion of anti-inflammatory cytokines, interleukin-2 neutralization, and direct suppression of effector T cells. Graves’ disease (GD) is a thyroid-specific autoimmune disorder primarily attributed to the breakdown of tolerance to the thyroid-stimulating hormone receptor. Given the limitations of currently available GD treatments, identifying potential pathogenetic factors for pharmacological targeting is of paramount importance. Both functional impairment and frequency reduction of Tregs seem likely in GD pathogenesis. Genome-wide association studies in GD have identified polymorphisms of genes involved in Tregs’ functions, such as CD25 (interleukin 2 receptor), and Forkhead box protein P3 (FOXP3). Clinical studies have reported both functional impairment and a reduction in Treg frequency or suppressive actions in GD, although their precise involvement remains a subject of debate. This review begins with an overview of Treg phenotype and functions, subsequently delves into the pathophysiology of GD and into the existing literature concerning the role of Tregs and the balance between Tregs and T helper 17 cells in GD, and finally explores the ongoing studies on target therapies for GD.

## 1. Introduction 

T regulatory cells (Tregs) constitute a subset of CD4^+^ T lymphocytes and serve as negative regulators of inflammation impacting various conditions including autoimmunity, infections, cancer, antimicrobial resistance, allergy, and transplantation [1,2,3,4,5,6,7,8,9]. Tregs’ primary function is to preserve self-tolerance and protect against autoimmune disorders [3]. Recent research has comprehensively profiled Treg markers, shedding light on the mechanisms involved in Treg function and plasticity. Results suggest that any compromise in Treg phenotype, proliferation, and adaptability to microenvironment is potentially implicated in the development of aberrant immune response against self-antigens and autoimmunity [3,4]. Autoimmune diseases encompass a wide spectrum of conditions that differ in their nature, origin, pathology, and prognosis. While genetic predisposition plays a role, environmental factors, such as infections, toxins, hormonal changes, and dietary factors, also contribute to disease development. Ongoing research aimed at unraveling the underlying mechanisms of these conditions, discovering new therapeutic targets, and advancing precision medicine approaches holds the promise of improved management and outcomes for those living with autoimmune diseases. Autoimmune thyroid diseases (AITDs), which include Graves’ disease (GD) and chronic autoimmune thyroiditis, are the most frequent organ-specific autoimmune disorders. AITDs are characterized by an aberrant immune response towards thyroid antigens (thyroglobulin, thyroid peroxidase, and thyroid-stimulating hormone receptor), immune infiltration of the thyroid gland, and circulating autoantibodies. Numerous studies have evaluated the role of Tregs in AITDs, yet these investigations have yielded conflicting results, primarily due to the absence of reliable markers to classify Tregs [5,6]. Two key reasons make knowing Treg function crucial: the potential to enable early identification of subjects prone to developing autoimmunity and the identification of novel immunomodulatory therapies with the potential for substantial public health and economic benefits. In addition to providing an immunological overview of the pathogenesis of GD focusing on Treg and Th17 cells, this review provides a synthetic description of future treatment options for GD.

## 2. Methods

The aim of this narrative review was to examine the current literature on Treg cells in GD encompassing their involvement in the early stages and progression of the disease and to propose potential therapeutic strategies involving their modulation. Original articles, systematic reviews, and meta-analyses in the English scientific literature were retrieved from inception to March 2023 using the following terms: Graves’ disease, Graves’ disease pathogenesis, autoimmune thyroid disorders, Graves’ disease mouse models, T regulatory cells/Treg and autoimmunity, T regulatory cells/Treg in myxedema, T regulatory cells/Treg in animal studies, T regulatory cells/Treg and autoimmune thyroid disorders, T helper 17/Th17, and Graves’ disease. Exclusion criteria were articles not focusing on Treg, articles reporting duplicate results or written in a language different than English, and those with results not clearly deducible from the text. Fifteen research articles encompassed the inclusion and exclusion criteria for human studies on the main topic (GD and Treg) and are reported in table in Section 3.6. This review initiates with an exploration of Treg phenotype and functions, followed by an in-depth analysis of the pathophysiology of GD. It then delves into the existing literature concerning the interplay between Tregs and T helper 17 cells in GD. Finally, the review explores the current ongoing research on targeted therapies for GD, along with the potential applications of Treg cell therapy in the context of GD.

## 3. Results and Discussion

### 3.1. Regulatory T Cell (Treg)

In a state of physiological equilibrium, the immune system is anticipated to promptly react to foreign antigens by triggering a highly effective inflammatory response while concurrently retaining the capacity to modulate and attenuate this response through anti-inflammatory mechanisms. One important mechanism involved in autoimmunity diseases is related to the phenomenon of molecular mimicry regarding antigen specificity of T or B cells towards infectious agents cross-reacting with self-peptides in susceptible individuals.

Perturbations in this homeostatic equilibrium can lead to the onset of autoimmune diseases and chronic inflammation when an excessive immune system response occurs or to the persistence of infections and immunodeficiency in cases of anergy.

Tregs are pivotal in this process since they act as negative regulators of inflammation in various conditions and preserve self-tolerance [1,2,3,4,5,6,7,8,9]. The most studied Treg subset is characterized by normal to high expression of the CD4, interleukin (IL)-2 receptor alpha chain (CD25 or IL-2R), and Forkhead box protein P3 (FoxP3), and low levels of CD127 (IL-7 receptor) [10,11,12,13,14]. CD25 and FoxP3 molecules are required for Tregs’ development, function, and stability [12,13,14,15,16]. Mutations of the FOXP3 gene have been related to the autoimmune syndrome “Immuno-dysregulation-polyendocrinopathy enteropathy X-linked” (IPEX), which is characterized by severe Treg dysfunction [14].

An alternative subset of Tregs is called IL10-producing type 1 regulatory (Tr1) cells [16]. Tr1 cells are characterized by the expression of CD49b and lymphocyte activation gene-3 (LAG-3) as specific biomarkers. Tr1 cells exert suppressive activity primarily via the secretion of IL-10. A CD4^+^ Treg cell subset characterized by a constitutive expression of CD69 but without Foxp3 expression was recently identified. These CD4^+^CD69^+^ Tregs mainly exert their suppressive activity through the production of IL-10 and transforming growth factor beta TGF-β [17].

Various subsets of CD8^+^ Tregs have been identified based on their expression of specific markers, including FoxP3, CD25, CD39, CD28, CD127, LAG-3, and CTLA-4 [18]. The study of CD8^+^ Treg cells is still evolving. Better knowledge of identification strategies, regulatory mechanisms, and induction pathways will enable researchers to discern the proportion of CD8^+^ Tregs relative to CD4^+^ T effector lymphocytes.

Treg cells variably express other markers, such as GITR (glucocorticoid-inducible TNF receptor), CTLA-4 (cytotoxic T lymphocyte-associated antigen), and growth arrest and damage-inducible proteins (Gadd45 α and β) [19], whose functional mechanisms are, in some cases, just partially defined. It is crucial to highlight that the variation in Treg phenotype is influenced by the surrounding microenvironment and controlled conditions [20].

In addition to being classified on marker expression, Tregs can be divided into two groups based on their developmental origin. Thymic Treg cells that originate from the thymus from the CD4 single-positive thymocytes are also known as natural or thymic Tregs (nTregs). nTregs leave the thymus as FoxP3^+^ cells enriched with T cell receptors (TCR) with a high affinity for self-peptides. nTregs are characterized by the expression of CD4, CD25, GITR, and FoxP3, are IL-2 dependent, and are anergic in the presence of antigen [21]. The induced Tregs (iTregs) develop in the periphery from conventional CD4^+^ T cells (CD4^+^FoxP3^−^CD25^low^ cells) after antigen encounter and in the presence of specific factors, such as TGF-β, IL-2, and all-trans retinoic acid. iTregs express CD4, CD25, CTLA-4, GITR, and FoxP3 [18,19,20,21,22,23,24,25].

The primary function of Tregs is to maintain immunologic self-tolerance, either directly via interaction with an antigen-specific target cell or indirectly via bystander suppression [26], in which Tregs specific for one antigen also suppress immune responses against other surrounding antigens. Tregs act via specific cell surface receptors, inhibitory cytokines (IL-10, TGF-β, and IL-35) release [27], and IL-2 capture [28]. When peptide-loaded MHC-II on antigen-presenting cells (APCs) binds to the T cell receptor (TCR) complex on Tregs, a signal transduction cascade ensues, resulting in Tregs’ activation.

Under specific circumstances, such as autoimmune disorders and cancer, Tregs respond to unusual environmental stimuli and display altered stability, plasticity, and tissue-specific heterogeneity, resulting in context-dependent suppressive activities. Although Tregs usually have an anti-inflammatory FOXP3^+^ phenotype, they can rarely acquire proinflammatory T helper (h)1-like or Th17-like phenotypes by the expression, respectively, of T-bet or STAT3 [27,28,29], or they can upregulate the expression of IFN regulatory factor-4 (IRF4) and Gata-3, thus, acquiring a Th2-like phenotype [30,31].

### 3.2. Regulatory T Cells in Autoimmunity

Autoimmune diseases represent a heterogeneous group of over 80 chronic disorders in humans that collectively impact a significant portion of the global population with a vast array of clinical manifestations, ranging from localized symptoms to systemic and multiorgan involvement. This heterogeneous group of disorders shares the breakdown of self-tolerance against endogenous proteins, leading to immune-mediated tissue and organ damage. Numerous studies have highlighted that Treg dysfunction is a common denominator in autoimmunity [32,33]. Reduced frequencies and impaired suppressive functions of Tregs have been identified in a wide spectrum of autoimmune diseases, including multiple sclerosis, rheumatoid arthritis, type 1 diabetes mellitus, systemic lupus erythematosus, inflammatory bowel disease, autoimmune hepatitis, Guillain–Barré syndrome, and Crohn’s disease [32]. This review will specifically focus on GD.

Clarifying the mechanisms of Treg actions is crucial in the lifelong maintenance of immune self-tolerance and homeostasis [34]. The main challenge in the search for efficacious therapies is to eliminate self-reactivity while maintaining the immune system’s capacity to protect against pathogens [35]. Many of the available therapies for autoimmune diseases do not achieve this delicate balance since they broadly suppress immune system function (e.g., corticosteroids) or target specific cytokines (e.g., TNF-α, IL-6) and cytokine receptors.

### 3.3. Graves’ Disease

Graves’ disease (GD) is a thyroid-specific autoimmune disorder, which causes hyperthyroidism and goiter [36]. GD is more frequent in females than males and may coexist with other autoimmune diseases in the so-called polyglandular autoimmune syndrome (PGA) [37]. Signs and symptoms of GD are related to the effect of elevated thyroid hormone (thyroxine and triiodothyronine) levels on target organs and increased metabolic rate and include fatigue, myalgias, fine tremor, palpitations and arrhythmias, weight loss, and heat intolerance [36]. The severity of symptoms depends on both the grade of hyperthyroidism and the demographic and clinical features of the patients. Hyperthyroidism may be isolated or associated with extrathyroidal manifestations, the most frequent being Graves’ orbitopathy (GO), which occurs in 25–30% of cases [38,39,40]. GO is characterized by exophthalmos and impaired eye motility resulting from inflammation of soft orbital tissue, increased adipogenesis, elevated glycosaminoglycan synthesis, edema in extraocular muscles, and fibrosis loss of tolerance to the thyroid-stimulating hormone receptor (TSH-R) and insulin-like growth factor-1 receptor (IGF-1R), which are overexpressed in orbital tissue affected by GO and are the primary factor in GO pathogenesis [39]. Given the colocalization of TSH-R and IGF-1R in ocular fibroblasts and thyrocytes, the IGF-1R signaling pathway may be activated either directly by IGF-1R antibodies (IGF-1R-Ab) or indirectly by TSH-R antibodies (TSHR-Ab) through synergistic crosstalk between TSH-R and IGF-1-R. Rarer complications of GD include thyroid dermopathy (1.5% of cases) and thyroid acropachy (0.3% of cases) [38], whose pathogenesis has not been unveiled yet. These extrathyroidal manifestations are specific to GD hyperthyroidism and do not occur in nonautoimmune forms of hyperthyroidism, such as multinodular toxic goiter.

GD treatment is still imperfect since the only available pharmacological options are thionamide antithyroid drugs (ATDs), which target hormone synthesis but not the underpinning autoimmune process, with a high recurrence rate after withdrawal. Alternative treatments are radioactive iodine (RAI) and thyroidectomy, which inevitably cause permanent hypothyroidism [40]. Accumulating acquisitions on GD pathogenesis paved the way for novel treatment strategies that target specific mechanisms of the immune response, including self-antigen presentation to effector lymphocytes and B cell activation, but results are still preliminary [40]. The introduction in the clinical practice of these new treatment strategies will completely change patients’ management, increasing the chance of a permanent cure.

### 3.4. Graves’ Disease Pathogenesis

GD is ultimately due to the breakthrough of tolerance to the thyroid-stimulating hormone receptor (TSHR). TSHR-autoantibodies (TSHR-Ab) inappropriately stimulate the TSHR, causing excessive thyroid hormone release (hyperthyroidism) and thyroid cell growth (goiter) [41]. Genetic susceptibility includes restricted T cell receptor (TCR) repertoire diversity and polymorphisms of genes encoding for proteins involved in the immune process, such as HLA complex (genes coding HLA DRB1*0301 and DQA1*0501 in Caucasians), CTLA-4, CD25, CD40, protein tyrosine phosphatase nonreceptor type 22 (PTPN22), and ribonuclease T2 (RNASET2) [41,42]. TSHR, located on chromosome 14q31, is a specific susceptibility locus for GD, and in particular its extracellular domain is highly immunogenic. The onset of GD is associated with nongenetic variables, which encompass endogenous factors like pregnancy and the postpartum period, as well as environmental stimuli including stress, infection, nicotine, diet, and microbiota composition. In addition to the widely recognized role of iodine status in maintaining thyroid health, recent attention has been directed towards other micronutrients, including selenium and vitamin D, which have emerged as possible regulators of thyroid autoimmunity [43].

Failure of negative regulation of inflammation plays a pivotal role in GD pathogenesis. The thyroid gland of GD patients has characteristic lymphocytic infiltration of T and B lymphocytes that are autoreactive to the TSHR after evading thymic and peripheral deletion. Abnormal HLA-DR expression allows thyroid cells to directly present autoantigens to autoreactive CD4^+^ T helper cells, which activate, proliferate, and trigger TSHR-Abs-producing plasma cells [44,45]. Once exposed to a specific antigen, naïve CD4^+^ T cells can differentiate in proinflammatory (Th17) or immunosuppressive (Treg) subsets. Infiltrating Th17 cells play a key role by producing proinflammatory cytokines that, in turn, activate resident T cells. Innate immune cells have a dual role in the immune response, encompassing both antigen presentation to effector lymphocytes and promotion of inflammation by thyroid infiltration and cytokines release [46]. This is supported by several reports, including the expansion of proinflammatory monocytes and the reduced killing ability of natural killer (NK) cells in GD patients [46,47]. Recent findings suggested that thyroid cells release the protein RNASET2 at the onset of GD hyperthyroidism, which may alarm and recruit immune cells to mitigate inflammation [47]. In turn, most immune cells can secrete RNASET2.

### 3.5. Mouse Models of Graves’ Disease

Various mouse models have been developed to study GD [48]. In 1996, Shimojo et al. successfully induced, for the first time, the authentic Graves’ disease model in mice [49]. Their novel approach involved repeated injections of fibroblasts stably transfected with cDNAs for the human TSHR and the major histocompatibility complex (MHC) class II into syngeneic AKR/N mice. A similar approach was later reported, immunizing mice with a B cell line (M12 cells) that constitutively expresses endogenous MHC class II and is transfected with the TSHR (the M12-TSHR model). Other models involving different approaches were successfully established: (i) TSHR cDNA vaccination using a eukaryotic expression plasmid (the DNA-TSHR model); (ii) immunization with recombinant adenovirus coding the TSHR (Ad-TSHR) (the Ad-TSHR model); (iii) immunization with dendritic cells (DC) infected with Ad-TSHR (the DC-TSHR model); (iv) model involving combined DNA-TSHR and in vivo electroporation; and (v) a spontaneous and inheritable model generated by transgenic expression of the variable region of a monoclonal anti-TSHR antibody (B6B7) isolated from a Graves’ patient’s lymphocytes. Alternatively, GD has been induced via immunizing plasmids or recombinant adenovirus vectors expressing TSHR (especially the more immunogenic TSHR-289) [50,51,52,53,54,55].

The introduction of TSHR knockout (KO) mice has revealed that, at least in mice with this genetic background (C57BL10/129), central tolerance may not play a major role for the TSHR. Instead, under these conditions, peripheral tolerance emerges as crucial, with Tregs playing a central role. Specifically, TSHR KO mice lack central tolerance because the TSHR is no longer an autoantigen, eliminating the expectation that immunization with TSHR would be stronger in KO than in wild-type (wt) mice. This unexpected outcome highlights the potential role of Tregs in Graves’ disease pathology.

Some of these models failed to induce hyperthyroidism in DBA/2J mice, raising questions about the presumed influence of hereditary factors in mouse susceptibility to GD. The technique of electroporation was employed to augment the expression of human TSHR. Enhancement of disease severity in susceptible BALB/c mice and disease induction in resistant C57BL/6 mice were obtained when immunological modification resulted in the depletion of CD4^+^CD25^+^ T cells. These findings supported the hypothesis that an imbalance between Teff cells and Tregs may play a significant role in the development and progression of GD [52]. In susceptible mice, hyperthyroidism induction and lymphocytic infiltration of the thyroid gland were increased by reducing CD8^+^CD122^+^ and CD4^+^CD25^+^ Tregs [53]. In the murine model of GD induced by Ad-TSHR289 immunization, Zhou et al. found reduced proportions of Tregs in the Ad-TSHR289 group, while the number of Th17 cells and the expression of transcription factor RORγt mRNA was unaltered between the Ad-TSHR289 and Ad-control groups [56]. These findings suggest that the development of GD in this model may be linked to the reduction of Tregs and an increased Th17/Treg ratio.

Yuan et al. used the same model for GD and reported that mice immunized with Ad-TSHR289 had a decreased frequency of Tregs and expression of FOXP3 mRNA as well, while the expression of IL-17 mRNA in the spleen was higher in the Ad-TSHR289 group [57]. The results from these two studies support the hypothesis that, in this animal model, GD may be associated with reduced Tregs and increased IL-17 gene expression that might not be necessarily associated with Th17 cells but with other IL-17-producing cells.

### 3.6. Regulatory T Cells in Graves’ Disease

Changes in Treg function or frequency are expected to have an impact on the initiation and course of GD (Figure 1) [58,59,60]. Possible susceptibility genes are FOXP3, CTLA-4, and CD25 [58,59,60], whose polymorphisms might impair Treg inhibitory functions or proliferation [61,62]. The -3279AA genotype of the FOXP3 gene, which leads to defective transcription of the gene, was recognized in 11.3% of GD patients with persistent hyperthyroidism and was absent in those in remission [62]. Single nucleotide polymorphisms (SNP) rs3761548 and rs3761549 of the FOXP3 gene have been associated with increased GD susceptibility, while the 3279CA SNP has been associated with a higher remission rate [62].

Clinical investigations provided contradictory outcomes, with some studies reporting no changes in Treg count [63,64,65], while others described either reduced or increased Treg frequency [66,67] (Table 1).

Bossowski et al. found considerably lower rates of CD4^+^FoxP3^+^ and CD4^+^CD25^high^ cells in untreated GD patients compared to healthy controls (HC) and a statistically significant correlation between TSHR-Ab levels and the percentage of CD4^+^IL-17^+^/CD4^+^CD25^+^CD127^+^, CD4^+^IL-17^+^/CD4^+^CD25^+^CD127 FOXP3^+^ T cells [68]. Comparable results were obtained by Liu and coworkers, who also observed increased Th17 and Th1 frequencies [69]. Similarly, Mao et al. reported a lower proportion of CD4^+^CD25^+^Foxp3^+^ Tregs in the peripheral blood of untreated and euthyroid GD patients and a negative correlation between TSHR-Ab levels and the frequency of CD4^+^CD25^+^FoxP3^+^ Tregs in untreated GD patients [65]. Further functional studies revealed that Treg suppressive ability was reduced in GD patients due to the influence of pathological dendritic cells and high thyroid hormone levels [65]. In another study, Treg’s number was lower in TSHR-Ab^+^ GD patients compared to patients in remission (due to ATD or RAI) and HC [70]. Accordingly, Chen et al. identified significantly diminished Treg frequency in patients with TSHR-Ab persistence despite long-term ATD treatment compared to remittent GD and HC [71]. RNA-seq and flow cytometric analysis revealed the upregulation of genes coding for proinflammatory cytokines and chemokines and the downregulation of Treg-related genes, such as SATB1, STAT5A, and TNFRSF18 [71]. During the follow-up of young GD patient responders to ATDs, Tregs’ frequency was restored to levels of HC [72]. Taken together, these findings suggest that Tregs’ depletion may be associated with the production/persistence of TSHR-Ab. According to the pathogenetic hypothesis of weakened Treg functions, Wang et al. observed that levels of IL-10, which is anti-inflammatory, and the expression of FOXP3 were significantly lower in GD patients compared to HC [63]. Conversely, Pan et al. observed higher levels of expression of message (m)RNA for Gadd45α, Gadd45 β, GITR, and CD25 and no changes in FOXP3 levels in both untreated and in-remission GD patients compared to HC, suggesting a compensatory Treg response to quench inflammation [64]. Increased circulating Tregs’ frequency with impaired suppressive functions was identified by another research group, which gathered Tregs such as CD69^+^NKG2D^+^IL-10^+^ or CD69^+^, GITR^+^, FoxP3^+^, IL-10^+^, and TGF-β^+^ [73,74,75]. Coculturing Tregs from 7 GD patients with blood-derived effector T cells, Glick et al. demonstrated that Tregs from GD patients had an impaired ability to suppress effector T cells, lower secretion of IL-10 and IL-2 and higher IL-17A production compared to HC [76]. Previous studies revealed that IL-17A production by Tregs was related to reduced suppressive activity [77,78,79,80].

Clearer insights came from a meta-analysis of six studies revealing a reduced frequency of circulating Tregs (identified as CD4^+^CD45RA^+^, CD4^+^CD25^+/high^, CD4^+^CD25^+^ FOXP3^+/high^, CD4^+^CD25^+^CD69^−^) in untreated GD patients (sample size 258 patients) compared to matched HC [81]. These findings were stronger in the studies in which Tregs were gathered as CD4^+^CD25^+^FoxP3^+/high^ [81], as confirmed by another recent meta-analysis [66], despite high heterogeneity. FOXP3 mRNA levels in peripheral blood mononuclear cells were significantly lower in GD patients in a subanalysis of six studies even if the pathogenetic role of FOXP3 *rs3761548* SNP could not be confirmed [82]. No differences emerged in GD patients in remission, but no conclusion could be inferred about Tregs’ changes during hyperthyroidism treatment due to the small sample of patients prospectively followed [81].

Since circulating Tregs may not always correspond to intrathyroidal infiltrate, few research studies have focused on this issue. Although several studies [64,68,75] observed higher recruitment of Tregs in inflamed thyroid tissue, those cells were unable to stop or diminish the ongoing inflammatory process. Nakano et al. observed that the proportion of intrathyroidal Tregs from 15 GD patients undergoing surgery was lower than the circulating proportion in the same patients and in HC due to increased Fas-mediated apoptosis [66]. These findings support the hypothesis regarding the failure of Tregs to infiltrate the thyroid gland due to dysfunction or apoptosis contributing to the development of the disease.

**Table 1 ijms-24-16432-t001:** Main clinical studies on T regulatory cells in Graves’ disease patients.

First Author, Year [Ref]	Main Findings in the Study in GD Patients Compared to HC
Marazuela et al., 2006 [75]	increased frequency of CD4^+^ lymphocytes expressing GITR, FoxP3, IL-10, TGF-β, and CD69 in GD patients prevalent thyroid infiltration by CD69^+^, CD25^+^, and GITR^+^ cells, with moderate levels of FoxP3^+^ lymphocytes defective suppressive Tregs’ function
Wang et al., 2006 [63]	no difference in the proportion of CD4^+^CD25^+^ T cells reduced secretion of IL-10 and expression of FoxP3
Nakano et al., 2007 [66]	decreased frequency of circulating and intrathyroidal CD4^+^CD25^+^ (both CD69^+^ and FoxP3^+^) cells increased frequency of apoptotic intrathyroidal CD4^+^ cells compared to peripheral CD4^+^ cells in AITD patients increased proportion of apoptotic intrathyroidal CD4^+^CD25^+^ cells compared to CD4^+^CD25^−^ cells
Fountoulakis et al., 2008 [83]	no differences in CD4^+^CD25^+^ and CD4^+^CD25^high^ cells’ frequency
Pan et al., 2009 [64]	no differences in CD4^+^, CD4^+^CD25^+^, and CD4^+^CD25^+(int−high)^CD127^+/low^ Tregs’ frequency/number in active GD patients vs. HC higher levels of mRNA for GADD45 alpha and beta, GITR, and CD25 in GD patients
Mao et al., 2011 [65]	decreased frequency of circulating CD4^+^CD25^+^FoxP3^+^ Tregs
Glick et al., 2013 [76]	no differences in the frequency of Tregs reduced ability to inhibit Teff cells’ proliferation
Bossowski et al., 2013 [68]	decreased frequency of circulating CD4^+^FoxP3^+^ and CD4^+^CD25^high^ T cells no differences in frequency of CD4^+^CD25^+^CD127^low^FoxP3^+^ T cells
Klatka et al., 2014 [72]	lower percentages and absolute counts of Treg cells at GD onset increase in the percentages and absolute counts of Treg lymphocytes during ATD
Rodríguez-Muñoz et al., 2015 [74]	circulating MVs from AITD patients have a functional role and can inhibit Treg differentiation
Rodríguez-Muñoz et al., 2016 [73]	increased percentage of circulating CD4^+^CD69^+^IL-10^+^, CD4^+^CD69^+^NKG2D^+^, and CD4^+^CD69^+^IL-10^+^NKG2D^+^ cells impaired ability of CD69^+^ Treg cells to suppress Teff cells
Li et al., 2016 [84]	lower frequency of circulating CD4^+^FoxP3^+^T cells lower levels of transcription factor FOXP3 mRNA elevated Th17/Treg ratio
Qin et al., 2017 [85]	lower frequency of circulating Tregs lower expression of FOXP3 gene elevated Th17/Treg ratio
Teniente-Serra et al., 2019 [70]	lower frequency of activated Treg cells (CD3^+^CD4^+^CD25^high^CD127^–/low^CCR4^+^CD45RO^+^HLA-DR^+^) percentage of Treg and activated Treg cells significantly lower in untreated GD patients compared to GD patients in remission
Chen et al., 2020 [71]	lower frequency of circulating Tregs downregulation of Treg-related genes, such as SATB1, STAT5A, and TNFRSF18 no changes in expression levels of FOXP3

Abbreviations: glucocorticoid inducible TNF receptor—GITR; Forkhead box protein P3—Foxp3; Interleukin—IL; Transforming growth factor beta—TGF-β; Graves’ disease—GD; healthy controls—HC; Growth arrest and damage-inducible proteins alpha—Gadd45 alpha; Antithyroid drugs—ATD; Microvesicles—MVs; Autoimmune thyroid disorders—AITD; T helper cells—Th. The representation of specific markers is as follows: “+” indicates the expression of the marker; “−” denotes no expression of the marker; “high” signifies high expression of the marker on the cells; “int” represents intermediate intensity of the marker expression; and “low” indicates that the marker is expressed on the cells but at a low intensity.

### 3.7. T Regulatory and T Helper 17 Cells’ Interplay in Graves’ Disease

The transition of Tregs toward proinflammatory subsets (Th17 and Th1 lymphocytes) may be one of the causes of their impaired antiphlogistic action in GD, which might contribute to the perpetuation of the autoimmune process [86,87,88,89,90,91]. Consistently, increased frequencies of FOXP3^+^/IL-17^+^ and FOXP3^+^/IFN-γ^+^ cells emerged in several autoimmune diseases [92]. Li et al. observed that in GD the percentage of CD4^+^IL-17^+^ T cells and the expression of RAR-related orphan receptor gamma (RORγt, proinflammatory) mRNA were increased, while CD4^+^FOXP3^+^ Treg cells and the mRNA level of FOXP3 were reduced [84]. Nanba et al. reported a higher percentage of peripheral Th17 cells in patients with intractable GD [93]. These results were confirmed by the extensive meta-analysis by Chen et al., which included 1302 newly diagnosed GD patients and 1815 HC [82].

It can be hypothesized that some biological molecules may exert immune modulation by regulating the Th17/Treg cell ratio [94]. Microvesicles (MVs) isolated from patients with autoimmune thyroid disorders were unable to inhibit the differentiation of FOXP3^+^ Tregs while stimulating the proliferation of Th17 cells in vitro [73]. The frequency of Th17 expressing the costimulatory receptor called signaling lymphocytic activation molecule family 1 (SLAMF1) was higher, while Treg cells expressing SLAMF1 were diminished in GD. Tregs that had reduced expression of SLAMF1 were dysfunctional [95].

Even though the Th17/Treg cell ratio can be a useful marker for assessing the severity of disease in animal models and human diseases [96], the extent of Treg cell conversion into Th17 and Th1 lymphocytes in GD remains to be unveiled.

### 3.8. New Treatment Strategies for Graves’ Disease

Thionamide antithyroid drugs (ATD) currently represent the first-line approach and the sole pharmacological option for GD treatment [40]. ATD action is based on reducing thyroid hormone synthesis via the inhibition of thyroid peroxidase. Some immunosuppressive effects of ATD have emerged in a few translational studies [97], but whether this was a direct effect of the drug or the result restoration of euthyroidism remains to be clarified [40,98]. Thionamides are quite safe, with rare serious adverse events. However, their efficacy is limited by a high rate of relapse after withdrawal [40]. Targeted therapy represents a potentially effective approach to GD treatment [40,99] (Figure 2). Available experimental drugs address several stages of the immune response, either specifically targeting TSHR/TSHR-Ab or nonspecifically reinstating self-tolerance [40]. The first group includes (a) ATX-GD-59, which is a small synthetic peptide built with two TSHR immunodominant epitopes (antigen-processing independent epitopes) which bind with high affinity to HLA-DR on the surface of immature dendritic cells into draining lymph nodes and induce tolerization; and (b) TSHR-blocking antibodies (ANTAG-3, VAK-14, S37a) or small molecules antagonizing TSHR-Ab signaling (K1-70). These drugs have shown good results in preliminary or preclinical studies. The second category encompasses pharmaceutical agents that have been previously employed in the treatment of other autoimmune disorders; for instance, rituximab specifically targets the production of antibodies by B cells through the inhibition of CD20, while belimumab blocks the B-cell activating factor (BAFF). There are drugs currently undergoing advanced stages of experimental investigation, like iscalimab, which has been developed to impede the interaction between CD40 (a tumor necrosis factor receptor) and CD154 (present on activated T lymphocytes) [40].

### 3.9. Regulatory T Cells as Target Therapy for Autoimmune Disorders: Brief Insights

Treatment of autoimmune disorders is still challenging due to complex pathogenies and variable clinical presentations. Conventional treatments (e.g., steroids) for most autoimmune disorders are based on suppressing the immune response to modulate uncontrolled inflammation. Consequences of nonselective immune suppression include an increased risk of infection. A more precise approach, based on novel technologies (e.g., cellular engineering) might overcome this limitation and reduce treatment risks.

Recent progress in comprehending the pathogenesis of diseases and advancements in drug manufacturing techniques has resulted in the extensive adoption of targeted immunotherapy for treating autoimmune diseases.

Preliminary evidence suggests that treatments designed to augment either directly or indirectly the suppressive functions and number of Tregs show both safety and effectiveness [100]. Among the prevailing strategies, adoptive Treg transfer stands as the most widely employed approach in experimental investigations across conditions such as type 1 diabetes, systemic lupus erythematosus, multiple sclerosis, myasthenia gravis, autoimmune hepatitis, irritable bowel syndrome, and Guillain–Barré syndrome, among others [101,102]. Despite promising outcomes, the utilization of autologous Treg therapy faces notable biological, technological, and economic hurdles [100].

Selection and isolation of Treg cells necessitate careful consideration of factors such as their maturation status, source (thymic, resident, peripheral blood, or umbilical cord blood), and markers (such as FoxP3, CD25, TGF-β). To reach an optimal Treg concentration, several post enrichment strategies have been investigated involving the addition of IL-2, rapamycin, TGF-β, and trans-retinoic acid [32,100,101]. In response to these challenges, ongoing research endeavors are exploring the application of genetically engineered Tregs. Additionally, diverse preclinical approaches aim to stimulate in vivo expansion or site-specific recruitment of Tregs, utilizing stimulating factors such as IL-2, IL-35, IL-27, IL-10, and CXCL11 or localized injections of CCL17 [102].

### 3.10. Regulatory T Cell-Based Strategies in Graves’ Disease

To our knowledge, there are no ongoing trials for Treg-based therapies in GD. It is crucial to preliminarily establish whether depletion, dysfunction, or site-specific recruitment of Tregs assume a central role in GD onset/progression and to identify the causative factors among genetic predisposition, environmental triggers, or elevated levels of thyroid hormones. Without this foundational knowledge, experiments may lack specificity and fail to address the root causes of the disease.

Nonpharmacological approaches have been explored to improve Treg actions in AITD. Xue et al. demonstrated that a selenium-enriched diet upregulated Treg expansion and reduced thyroid tissue inflammation in mice with iodine-induced thyroiditis [103]. Similarly, administration of calcitriol resulted in the expansion of Tregs, downregulation of IL-17, and improvement of Treg suppressive effectiveness in vitro [104]. In a recent trial, the addition of selenium and cholecalciferol to ATD provided a prompter resolution of hyperthyroidism and an increase in Treg frequency in GD hyperthyroidism of recent onset [43].

## 4. Conclusions

The balance between immune activation and regulation is a key determinant in the pathophysiology of GD. Dysregulation of Tregs, whether due to their depletion, impaired function, or defective site-specific recruitment, appears to play a pivotal role in GD pathology, but controversies have emerged in the available literature. Patient heterogeneity, variations in Treg markers employed for analysis, methodological discrepancies, and relatively small sample sizes might explain this controversy. While a more comprehensive characterization of Treg cells in GD is critical, existing research predominantly suggests an impairment in their suppressive efficacy, stemming from reduced Treg percentages or dysfunction [83,84].

Several factors might concur, including genetic polymorphisms of FOXP3 or CD25, increased mRNA transcription of proinflammatory molecules, decreased transcription of anti-inflammatory molecules, the direct impact of hyperthyroidism itself, and a thyroid inflammatory microenvironment (Figure 1).

Tregs impairment might favor the onset of GD, its clinical persistence, and the scarce response to ATD treatment. Whether Tregs’ dysfunction is implicated in the extra-thyroidal manifestation of GD as Graves orbitopathy and thyroid dermopathy remains to be clarified. Unfortunately, ATDs fail to achieve a long-lasting remission in almost half of cases. As our understanding of GD pathogenesis rapidly advances, numerous biological drugs targeting various stages of the immune response have been developed and are currently under study. Therapies based on the modulation of Treg functions have been explored in other autoimmune disorders with encouraging results but have not been investigated in GD yet.

If the role of Treg cells is better defined and detailed, targeting Tregs may emerge as a prospective strategy for ameliorating or even curing hyperthyroidism, along with its associated extrathyroidal effects. This avenue of research holds promise for enhancing therapeutic approaches to GD and its multifaceted clinical presentations.

Key points:Loss of central and peripheral immune tolerance is crucial in GD pathogenesis;Tregs mediate immune tolerance, acting as negative regulators of inflammation;Reduced Tregs’ number/function impairs immunoregulation and exposes susceptible subjects to autoimmunity development/propagation;Targeting Treg may emerge as a prospective strategy for ameliorating or even curing hyperthyroidism;Diverse markers characterize human Tregs, posing a challenge to the comparability of studies in autoimmune patients.

## Figures and Tables

**Figure 1 ijms-24-16432-f001:**
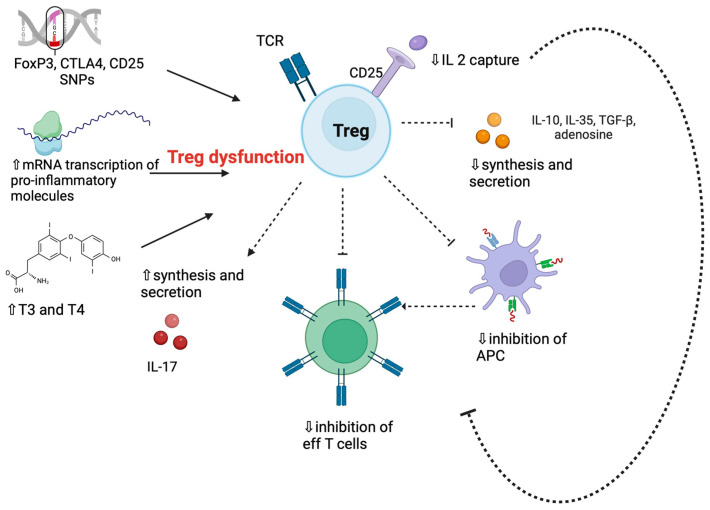
Potential mechanisms of T regulatory cell dysfunction in Graves’ disease. Abbreviations: APC, antigen-presenting cell; CTLA-4, cytotoxic T lymphocyte-associated antigen; Eff T cell, effector T cells; Forkhead box protein P3—FOXP3; IL, interleukin; T3 and T4, thyroid hormones; TCR, T cell receptor; Treg, T regulatory cells; ⇓, lower; ⇑, higher. Original illustration realized with Biorender.

**Figure 2 ijms-24-16432-f002:**
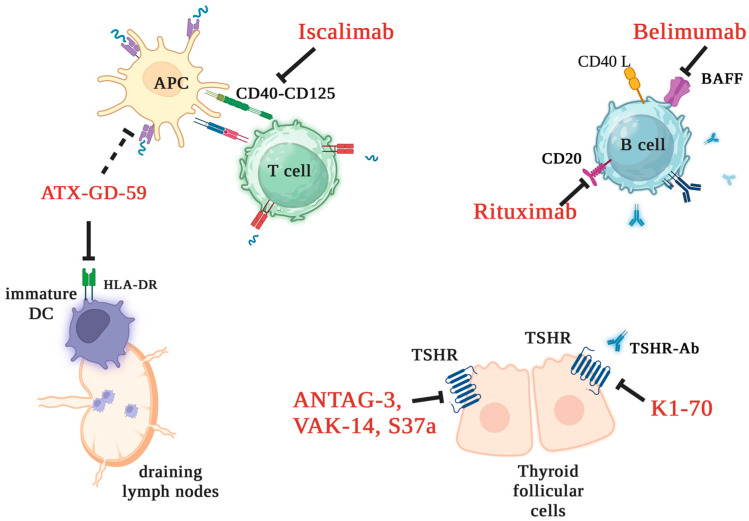
Target therapies in Graves’ disease. Abbreviations: APC, antigen-presenting cell; BAFF, B-cell activating factor; DC, dendritic cells; TSHR, thyroid-stimulating hormone receptor TSHR-Ab, antibodies to thyroid-stimulating hormone receptor. Original illustration realized with Biorender.

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
