# Peer review of "Regulatory T Cells in the Pathogenesis of Graves’ Disease"

_ijms, 2023, doi:10.3390/ijms242216432_

Round 1

Reviewer 1 Report

Comments and Suggestions for Authors

The authors produced a well written description of the overall issue (tregs and autoimmunity), then provided details as it pertains to Graves' disease. Future prospects of Treg therapy are discussed in light of analogous therapies used in other autoimmune diseases. Since there are no clinical trials yet, this review is relevant and timely.

Introduction
CD4 is the obvious focus of the review, however, CD8+ T regs are also known for a long time (authors mention briefly later on line 224). Some more information on the T reg subsets would be helpful in the introduction. For example are the subsets based on phenotype, function, cytokines, or lineage.
Do the subsets in question refer to induced T regs, or  natural T regs? (later on this is explained, so perhaps a bit more information early on in the article would help prepare the reader).

Main Text.
Around line 81, the concept of antigen specificity is also important, a foreign antigen can cause molecular mimicry, or, a T cell can react against self-antigen resulting in chronic autoimmune disease. Tregs keep threshold of activation high to minimize such interactions.

minor issues:

The meaning of 'peculiarly' on line 97 is not clear. Does it mean 'oddly' or 'uniquely'.

typo line 107 cel,l
line 112 activation of T regs (or T regs' activation)
line 119 should use GD acronym
Line 215 not clear which reference this result was from
line 231,240 etc  gene names uppercase FOXP3
line 280 TNF- the symbol is not coming out right
line 283 is IL-17 referring to IL-17A or one of the other ones.
Line 287, not sure what is high/+ , on the next line one of them is +/high, perhaps explaining the notation briefly would help reader interpret.
Line 363 multiple sclerosis should be lower case, and the other ones not named after a person.

Comments on the Quality of English Language

Is fine, just a few typos

Author Response

Response to Reviewers

Dear Editor,

we hereby resubmit the paper “Regulatory T cells in the pathogenesis of Graves’ disease”, which previously received a decision of minor revision. We appreciated the benevolent comments of the Reviewers and their professional suggestions. We revised the manuscript according to the Reviewers’ criticisms. As asked by Reviewer 1 and Reviewer 2 we changed, modified the text and added a list of key points summarizing the main findings of this narrative review. Our rebuttal is detailed in the file below. Changes to the original text are marked up by yellow color.

Reviewer 1

We thank Reviewer 1 for Her/His positive comments and for the minor requests for clarification along the text that we think could improve our article even further. All criticisms have been addressed in our revised version and are detailed below:

1) Introduction CD4 is the obvious focus of the review, however, CD8+ T regs are also known for a long time (authors mention briefly later on line 224). Some more information on the T reg subsets would be helpful in the introduction. For example, are the subsets based on phenotype, function, cytokines, or lineage.

Do the subsets in question refer to induced T regs, or natural T regs? (later on, this is explained, so perhaps a bit more information early on in the article would help prepare the reader).

  1. R) To address the valuable feedback, the following text includes additional information, particularly concerning CD8 T cell and its subsets. Additionally, the paragraph concerning specific CD4 subsets has been rephrased to enhance clarity and incorporate additional details regarding the lineage and function of specific subsets as described below with a new reference, #18: 18. Vieyra-Lobato, M.R. et al. Description of CD8+ Regulatory T Lymphocytes and Their Specific Intervention in Graft-versus-Host and Infectious Diseases, Autoimmunity, and Cancer. J. Immunol .Res. 2018, 2018, 3758713:

Another CD4+ Treg cell subset was recently identified, and it is characterized by a con-stitutive expression of CD69 but without Foxp3 expression. These CD4+CD69+ Tregs mainly exert their suppressive activity through the production of IL-10 and trans-forming growth factor beta TGF-β [17]. In addition, subset of Tregs that are expressing CD8 has been identified as well. In contrast to the well-established CD4+Foxp3+ Tregs, recent advances have begun to un-veil the biology of CD8+ Tregs. Various subsets of CD8+ Tregs have been identified based on their expression of specific markers, including FoxP3, CD25, CD39, CD28, CD127, LAG-3, and CTLA-4 (reviewed in [18]). The study of CD8+ Treg cells is still evolving. A comprehensive analysis of their identification, regulatory mechanisms, induction pathways, and other factors will enable researchers to discern the propor-tion of CD8+ Tregs relative to CD4+ T effector lymphocytes. Moreover, T regulatory cells express a variety of other markers with more variable expression, such as GITR (glucocorticoid-inducible TNF receptor), CTLA-4 (cytotoxic T lymphocyte-associated antigen), and growth arrest and damage-inducible proteins (Gadd45 α and β) [19]. It should be noted that many of these markers exhibit variable expression, and their functional mechanisms are still being intensively studied and, in some cases, remain ill-defined. It is crucial to highlight that the variation in Treg phenotype is influenced by the surrounding microenvironment and controlled conditions [20]. As stated above beside being classified based on the expression of the diverse markers Tregs can be divided into two groups based on their developmental origin. Thymic Treg cells that originate from the thymus from the CD4 single-positive thymocytes are usually called also known natural or thymic Tregs, nTregs. nTregs leave the thymus as FoxP3+ cells that are enriched for T cell receptors (TCR) with high affinity for self-peptides. Furthermore, they are characterized by expression of CD4, CD25, GITR and FoxP3, and are IL-2 dependent and are anergic in the presence of antigen [21]. The induced Tregs (iTregs) on the other hand, develop in the periphery from conventional CD4+ T cells (CD4+FoxP3-CD25low cells) after antigen encounter and in the presence of specific factors such as TGF-β, IL-2 and all-trans retinoic acid. iTregs express CD4, CD25, CTLA-4, GITR and FoxP3 [18-25].

2) Main Text. Around line 81, the concept of antigen specificity is also important, a foreign antigen can cause molecular mimicry, or, a T cell can react against self-antigen resulting in chronic autoimmune disease. Tregs keep threshold of activation high to minimize such interactions.

  1. R) We agree and fully share the importance of the molecular mimicry phenomenon in autoimmune diseases and have added a sentence introducing this key point, as follow: One important mechanism involved in autoimmunity diseases is related to the phenomenon of molecular mimicry regarding antigen specificity of T or B cells towards infectious agents cross-reacting with self-peptides in susceptible individuals.

Page 2, lines 85-90 of the revised version of the manuscript.

3) Minor issues: The meaning of 'peculiarly' on line 97 is not clear. Does it mean 'oddly' or 'uniquely'.

  1. R) We appreciate the reviewer's comment, as the use of the word "peculiarly" in this context could be subject to misinterpretation. While we lean towards the interpretation of "specifically," it's important to note that the expression of CD49b has been confirmed on other T cell types, as it plays a role in mediating the migration of T cells into extravascular spaces. In light of this, we have decided to entirely eliminate the word "peculiar."

4) typo line 107 cel,l

line 112 activation of T regs (or T regs' activation)

line 119 should use GD acronym

Line 215 not clear which reference this result was from

line 231,240 etc. gene names uppercase FOXP3

line 280 TNF- the symbol is not coming out right

line 283 is IL-17 referring to IL-17A or one of the other ones.

Line 287, not sure what is high/+, on the next line one of them is +/high, perhaps explaining the notation briefly would help reader interpret.

Line 363 multiple sclerosis should be lower case, and the other ones not named after a person.

  1. R) We have made all the necessary corrections following criticisms of the Reviewer 1, as indicated in the text, and in figure legend.

Reviewer 2 Report

Comments and Suggestions for Authors

hello

thank you for your interest review

the title, meets the paper's aim and topic,

graves Basedow is an interesting but sometimes challenging disease

the following chapters are well-structured and written

introduction is adequate

please write if there were any inclusion/exclusion criteria for a paper that you have reviewed, and how many of the reviewed papers meet the inclusion criteria to be used in your paper - and why?

a flow chart would be also good

Is Figure 1 author's own drawing? if so, please write - your own source under the figure

each paragraph seems to be well-organised

im missing some more clinical importances mentioned in the main text

same figure 2- is it your own drawing or is it copied from someplace?

im missing a chapter with some hints, and an explanation of this review for a normal clinician - what would be most interesting for him?

please add the top 5 highlighted bullet points at the end of the review

add some information why immunology and studying cell mechanisms is so important in today's medicine - is it evidence-based related?

I would highlight more efficiently top 3 most common and often seen complications of graves disease, and also write if there is any correlation with genes/immunologic status and the occurrence of some of those rare complications

conclusions are sufficient for the journal style

some minor grammar/style erros should be corrected

please improve the following

thank you 

Author Response

Reviewer 2

We thank Reviewer 2 for Her/His constructive comments which helped us to improve the manuscript quality. Criticisms have been addressed in this revised version of the manuscript and are detailed below:

  • “thank you for your interest review; the title, meets the paper's aim and topic, the following chapters are well-structured and written introduction is adequate”
  1. R) We thank the Reviewer for the nice comments and appreciation of our manuscript
  • please write if there were any inclusion/exclusion criteria for a paper that you have reviewed, and how many of the reviewed papers meet the inclusion criteria to be used in your paper - and why?
  1. R) This was a narrative Review and not a systematic review, thus a flowchart describing articles selection was not applicable. However, since we followed a strict method for articles selection we could clarify exclusion criteria in the text. We revised more than 4000 articles and selected 15 research articles addressing the topic of Treg in Graves’ disease in human. We also considered manuscript of basic/translational science since we aimed to report all available insights about this topic. To dive into this complex topic also a non-expert readers we decided to include literature exploring autoimmunity in general and available/on study innovative treatments strategies. The article has been modified accordingly, page 2 lines 72-75 of the revised version of the manuscript.

“ Exclusion criteria were: articles not focusing on Treg, reporting duplicate results or written in a language different than English, results not clearly deducible from the text. Fifteen research articles encompassed the inclusion and exclusion criteria for human studies on the main topic (GD and Treg) and were reported in Table 1.”

3) “Is Figure 1 author's own drawing? if so, please write - your own source under the figure”

“The same figure 2- is it your own drawing or is it copied from someplace?”

  1. R) Figures 1 and 2 are original and have been prepared with Biorender.com. Illustrations aimed to summarize the content of the article, to address the attention of the readers and to simplify mechanism of Treg actions/dysfunction (1) and available target therapies for GD (2). We added this phrase in the footnotes of Figures 1 and 2 “Original illustration realised withcom.

4) “im missing a chapter with some hints, and an explanation of this review for a normal clinician - what would be most interesting for him?”;

 “add some information why immunology and studying cell mechanisms is so important in today's medicine - is it evidence-based related?”

  1. R) We thank the Reviewer for Her/His comments. Current clinical practice lacks a pharmacological treatment able to definitively cure GD. The main patient concern about available pharmacological treatment (antithyroid drugs) is related to the risk of disease relapse, which is partially predictable. In case of a severe form of hyperthyroidism patient and clinician might decide to choose a definitive treatment, which, however, are burdened by permanent hypothyroidism

page 2, lines 58-62; page 4, lines 164-174 and page 11 lines 415-424 of the revised version of the manuscript.

“Two key reasons make knowing Treg function crucial: the potential to enable early identification of subjects prone to developing autoimmunity and the identification of novel immunomodulatory therapies with the potential for substantial public health and economic benefits. Besides providing an immunological overview of the pathogenesis of GD focusing on Treg and Th17 cells, this Review provides a synthetic description of future treatment options for GD.”

“GD treatment is still imperfect since the only available pharmacological options are thionamide antithyroid drugs (ATDs), which target hormone synthesis but not the underpinning autoimmune process, with a high recurrence rate after withdrawal. Alternative treatments are radioactive iodine (RAI) and thyroidectomy, which inevitably cause permanent hypothyroidism [40]. Accumulating acquisitions on GD pathogenesis paved the way for novel treatment strategies that target specific mechanisms of the immune response, including self-antigen presentation to effector lymphocytes and B cell activation, but results are still preliminary [40]. The introduction in the clinical practice of these new treatment strategies will completely change patients’ management, increasing the chance of a permanent cure.”

“Treatment of autoimmune disorders is still challenging due to complex pathogenies and variable clinical presentation. Conventional treatments (e.g. steroids) for most autoimmune disorders are based on suppressing the immune response to modulate uncontrolled inflammation. Consequences of non-selective immune suppression include an increased risk of infection. A more precise approach, based on novel technologies (e.g. , cellular engineering), might overcome this limitation and reduce treatment risks. Recent progress in comprehending the pathogenesis of diseases and advancements in drug manufacturing techniques has resulted in the extensive adoption of targeted immunotherapy for treating autoimmune diseases.”

5) Please add the top 5 highlighted bullet points at the end of the review

  1. R) We thank the Reviewer for her/his for this advantageous suggestion. We were pleased to create a list of key points (page 11, lines 488-496)

Key points

  • Loss of central and peripheral immune tolerance is crucial in GD pathogenesis.
  • Tregs mediate immune tolerance acting as negative regulators of inflammation.
  • Reduced Tregs number/function impairs immunoregulation and exposes to autoimmunity development/propagation.
  • Targeting Treg may emerge as a prospective strategy for ameliorating or even curing hyperthyroidism.
  • Diverse markers characterize human Tregs, posing a challenge to the comparability of studies in autoimmune patients.

6) I would highlight more efficiently top 3 most common and often seen complications of graves disease, and also write if there is any correlation with genes/immunologic status and the occurrence of some of those rare complications

  1. R) We thank the Reviewer for the suggestion. We added a brief description of key points the pathogenesis of Graves’ orbitopathy (GO) which is the most common and important extrathyroidal manifestation of GD (page 4, lines 154-162 of the revised version of the manuscript). Since GO is a very complex disease, and its pathogenesis has been just partially clarified an adequate description of the disease would require a long dissertation which is however outside the aim of this review. A relevant decrease in circulating CD4+CD25+CD127-cells frequency in patients with active GO emerged in some studies. Importantly, a reduction of circulating FoxP3+ Tregs and particularly of effector Tregs and an increased expression of CTLA-4 on naïve Tregs in subjects with both active and inactive ophthalmopathy have been described. In orbital tissue staining from GO patients, higher FoxP3 mRNA levels have been reported compared to healthy controls and in severe ophthalmopathy compared to milder forms. This suggests that Treg cells are sequestered in orbital tissue to overcome inflammation but further studies are ongoing.
